# Genetic Signatures for Distinguishing Chemo-Sensitive from Chemo-Resistant Responders in Prostate Cancer Patients

**Lemohang Gumenku [1], Mamello Sekhoacha [2,\*], Beynon Abrahams [3], Samson Mashele [1], Aubrey Shoko [4] and Ochuko L. Erukainure [5]**

[1] Department of Health Sciences, Central University of Technology, Bloemfontein 9300, South Africa; lgumenku@cut.ac.za (L.G.); smashele@cut.ac.za (S.M.)
[2] Department of Pharmacology, University of the Free State, Bloemfontein 9300, South Africa
[3] Department of Basic Medical Sciences, University of the Free State, Bloemfontein 9300, South Africa; abrahamsbr@ufs.ac.za
[4] Centre for Proteomics and Genomics Research, Cape Town 7925, South Africa; aubrey.shoko@cpgr.org.za
[5] Laser Research Center, University of Johannesburg, Doornfontein 2028, South Africa
[\*] Correspondence: sekhoachamp@ufs.ac.za

**Abstract:** Prostate cancer remains a significant public health concern in sub-Saharan Africa, particularly impacting South Africa with high mortality rates. Despite many years of extensive research and significant financial expenditure, there has yet to be a definitive solution to prostate cancer. It is not just individuals who vary in their response to treatment, but even different nodules within the same tumor exhibit unique transcriptome patterns. These distinctions extend beyond mere differences in gene expression levels to encompass the control and networking of individual genes. Escalating chemotherapy resistance in prostate cancer patients has prompted increased research into its underlying mechanisms. The heterogeneous nature of transcriptomic organization among men makes the pursuit of universal biomarkers and one-size-fits-all treatments impractical. This study delves into the expression of drug resistance-associated genes, ABCB1 and CYP1B1, in cancer cells. Employing bioinformatics, we explored the molecular pathways and cascades linked to drug resistance following upregulation of these genes. Samples were obtained from archived prostate cancer patient specimens through pre-treatment biopsies of two categories: good vs. poor responders, with cDNAs synthesized from isolated RNAs subjected to qPCR analysis. The results revealed increased ABCB1 and CYP1B1 expression in tumor samples of the poor responders. Gene enrichment and network analysis associated ABCB1 with ABC transporters and LncRNA-mediated therapeutic resistance (WP3672), while CYP1B1 was linked to ovarian steroidogenesis, tryptophan metabolism, steroid hormone biosynthesis, benzo(a)pyrene metabolism, the sulindac metabolic pathway, and the estrogen receptor pathway, which are associated with drug resistance. Both ABCB1 and CYP1B1 correlated with microRNAs in cancer and the Nuclear Receptors Meta-Pathway. STRING analysis predicted protein–protein interactions of ABCB1 and CYP1B1 with Glutathione S-transferase Pi, Catechol O-methyltransferase, UDP-glucuronosyltransferase 1-6, Leucine-rich Transmembrane and O-methyltransferase (LRTOMT), and Epoxide hydrolase 1, with scores of 0.973, 0.971, 0.966, 0.966, and 0.966, respectively. Furthermore, molecular docking analysis of the chemotherapy drug, docetaxel, with CYP1B1 and ABCB1 revealed robust molecular interactions, with binding energies of $-20.37$ and $-15.25$ Kcal/mol, respectively. These findings underscore the susceptibility of cancer patients to drug resistance due to increased ABCB1 and CYP1B1 expression in tumor samples from patients in the poor-responders category that affects associated molecular pathways. The potent molecular interactions of ABCB1 and CYP1B1 with docetaxel further emphasize the potential basis for chemotherapy resistance.

**Keywords:** prostate cancer; biomarker; chemotherapy resistance; proteomics; pharmacogenetics

## 1. Introduction

Prostate cancer (PC) is the most commonly diagnosed malignant tumor and a prominent contributor to cancer-related mortality in men [1]. In 2020, GLOBOCAN/IARC reported a staggering 1.4 million prostate cancer cases and 375 304 deaths worldwide, with prostate cancer emerging as the leading cause of cancer-related deaths in 46 countries, mainly in sub-Saharan Africa and Latin America [2]. Racial groups exhibit shared genomic variations that may influence cancer susceptibility. Africans experience a prostate cancer incidence approximately 2 to 3 times higher than Caucasians and Asians. However, the human genome is constantly exposed to both internal and external genotoxic stressors [3,4]. The gravity of the situation is underscored by the high incidence and limitations of available treatments. PC is marked by distressing clinical symptoms, including dysuria, oligospermia, hemospermia, and urinary complications, with aggressive progression and frequent metastasis [3]. Although chemotherapy is the primary option for managing hormone-resistant metastatic PC, alarmingly high treatment-failure rates have emerged due to drug resistance, necessitating urgent exploration of novel strategies [4].

Prostate cancer patients, despite sharing identical histology and comparable clinical parameters, present diverse molecular profiles. However, castration-resistant prostate cancer (CRPC) typically exhibits genomic instability, characterized by mutations in multiple genes, especially in advanced metastatic disease [5]. A recent genomic study examined sequencing data from both castration-sensitive and castration-resistant prostate cancer cases, revealing BRCA2 mutations as the most prevalent mutations, with a frequency of 12.7% in both cancer types [6]. The field of pharmacogenomics, which customizes treatments based on individual patient profiles and explores genetic impacts on drug reactions, holds significant potential for entirely transforming cancer treatment approaches.

Individual responses to treatment vary widely due to the interaction of multiple genes controlling drug response and toxicity. Genetic markers involved in drug transport and metabolism hold potential as more accurate predictors of therapeutic success. Genetic diversity, particularly through polymorphisms in drug resistance and metabolism genes, plays a pivotal role in determining individual responses to the same treatment regimen [7]. Despite international research harnessing the potential of genetic biomarkers for predicting drug responses in cancer treatment, South African researchers have yet to fully explore this therapeutic frontier.

ATP binding cassette (ABC) transporter proteins expressed in the plasma membrane are well-known contributors to multidrug resistance. These ATP-dependent transmembrane proteins function as efflux pumps and are part of the broader ABC transporter superfamily, enabling the transport of drugs and xenobiotics both into and out of cells [8]. Normal tissues host multidrug resistance protein (MRP) transporters, such as MRP2, MRP3, MRP4, and multidrug resistance (MDR)-1 protein (P-glycoprotein) [9]. In cancer cells, their overexpression is associated with increased efflux of chemotherapeutic drugs, contributing to the development of multidrug resistance [10]. Notably, the development of docetaxel resistance has been associated with relapse, with the upregulation of the MDR-1 gene encoding ABCB1 playing a crucial role [11].

Cytochrome (CYP) P450 constitutes a multi-gene superfamily of both constitutive and inducible heme-containing monooxygenases, actively participating in the metabolism of diverse xenobiotics and endogenous substrates [12]. Phase I drug-metabolizing enzymes identified in the human prostate encompass CYP1A2, CYP1B1, CYP2C19, CYP2D6, CYP3A5, and CYP4B1, which are present in both normal and tumorous tissue [9]. CYP4B1, which is responsible for activating arylamines through N-hydroxylation, poses an increased risk of bladder tumors and is associated with prostate cancer risk [13]. CYP1B1, an inducible member of the CYP450 superfamily, serves as a crucial tumor biomarker [14]. Overexpression of CYP1B1 has been documented in drug-resistant prostate cancer tumors. These findings collectively suggest that the upregulation of intratumoral steroidogenesis contributes to facilitating survival in CRPC in a castrate environment.

The present study aimed to address this gap by investigating the relationship between drug responses and potential genetic polymorphisms linked to drug transport and metabolism, thus establishing a comprehensive pharmacogenetic database predicting chemo-resistance specific to patients and offering crucial insights into possible treatment outcomes. This study focused on the ABC transport proteins MDR-1/ABCB1, which is critical for drug delivery, and the CYP450 metabolic enzyme CYP1B1, which impacts chemotherapy metabolism, clinical chemo-resistance, and overall treatment outcomes.

Recent advancements underscore the critical clinical significance of both germline genetic testing and somatic genomic profiling, particularly in cases of advanced prostate cancer. Furthermore, identifying mutations or pathogenic variants in germline genetics can have crucial implications for family members and prompt life-saving adjustments in healthcare management [15]. The landscape of genetic testing is evolving as more cancer susceptibility genes are discovered. While breast cancer testing practices have undergone refinement over nearly two decades, genetic testing for prostate cancer in clinical settings is still in its nascent stages [16].

The adoption of next-generation sequencing for tumor profiling is rapidly growing in oncology. Technological advancements have significantly reduced sequencing costs, facilitating the incorporation of multiple genes into a single panel test. Traditional single-gene testing has been cumbersome and time-consuming, with clinicians awaiting results from each test before proceeding [17]. Moreover, a negative outcome may imply that the relevant germline alteration remains unidentified. Consequently, multi-gene panel testing streamlines the diagnostic process, potentially uncovering additional germline alterations due to its broader genetic screening range. This comprehensive approach not only identifies key cancer-driving alterations but also aids in selecting therapies based on biomarkers [6].

Given the pressing healthcare concern, it is imperative to explore new avenues for enhancing PC treatment, especially in the advanced CRPC setting. While first-line chemotherapy with docetaxel has become standard, questions persist regarding the efficacy of prednisone alongside docetaxel [18]. Furthermore, the emergence of resistance to docetaxel highlights the need for predictive biomarkers to guide treatment decisions, an area where current clinical practice is deficient. This study seeks to address these critical issues, offering a novel approach to molecular strategies combating drug resistance in PC patients.

## 2. Materials and Methods

### 2.1. Archived Biopsy Samples

This study harnessed archived biopsy samples, specifically formalin-fixed paraffin-embedded (FFPE) tissues, sourced from the well-established clinical repository at Universitas Hospital, Bloemfontein, South Africa. Ethical approval for sample collection was obtained from the Health Science Research Ethics Committee, Provincial Health Research Committee, and the National Health Laboratory Service (Ethics Number: UFS-HSD2020/1278/2302-0004). Collected as a routine facet of medical practice before chemotherapy treatment, these biopsy specimens represent a valuable resource for exploring critical aspects of prostate cancer, offering insights into the disease's characteristics and progression. The samples were stratified into two distinct groups based on biopsy-derived sensitivity and treatment outcomes, categorizing individuals as either good responders (chemo-sensitive) or poor responders (chemo-resistant).

The requested specimens spanned records from 2016 to 2022. These tissues were meticulously collected in sterile nuclease-free tubes and subsequently transferred to the respective laboratories. Collection considerations encompassed variables such as age (patients' ages ranged between 40 and 80 years, with a median of 60 years), the administered treatment (docetaxel), clinical stage (IV), and clinical complications encountered. Clinical and pathological variables included the cancer stage, tumor type, and grade (Table 1). Table 2 below shows results for a total of 20 samples and a control.

**Table 1.** Clinical characteristics of prostate cancer patients.

| Patient Category | Symptomatic Improvement | Gleason Score | Progressive Disease | Serum PSA Level |
|---|---|---|---|---|
| Good Responders | | | | |
| 1 | Yes | 7 (4 + 3) | No | >5000 |
| 2 | Yes | 7 (4 + 3) | No | 7000 |
| 3 | Yes | 8 (4 + 4) | No | 300 |
| 4 | Yes | 7 (3 + 4) | No | >2000 |
| Poor Responders | | | | |
| 5 | No | 8 (4 + 4) | Yes | >2000 |
| 6 | No | 9 (4 + 5) | Yes | 209 |
| 7 | No | 7 (4 + 3) | Yes | 75 |
| 8 | No | 7 (4 + 3) | Yes | 103 |
| 9 | No | 7 (4 + 3) | Yes | 4513 |

**Table 2.** Mean Cq (quantification cycle) values (of 3 replicates) for each target and reference gene using preamplified cDNA samples as templates for qPCR.

| Sample Name | Biological Group | CqABCB1 | CqCYP1B1 | CqGAPDH | CqHPRT | CqHSPCB |
|---|---|---|---|---|---|---|
| 5N | Normal | 28.3 | 24.12 | 21.4 | 33.8 | 22.4 |
| 1N-1 | Normal | 30.1 | 23.86 | 17.6 | UND | 19.8 |
| 1N-2 | Normal | 32.8 | 28.92 | 27.2 | 28.2 | 24.0 |
| 6N | Normal | 27.8 | 22.93 | 17.8 | 27.7 | 19.7 |
| 7N | Normal | 26.1 | 20.38 | 15.6 | UND | 17.3 |
| 2N | Normal | 30.1 | 22.01 | 18.1 | 28.2 | 18.3 |
| 3N | Normal | 27.1 | 21.56 | 16.7 | 27.3 | 18.8 |
| 8N | Normal | 30.2 | 25.04 | 24.2 | 27.8 | 22.8 |
| 9N | Normal | UND | 30.42 | 29.1 | 25.4 | 26.5 |
| 4N | Normal | 29.4 | 24.80 | 20.7 | 24.8 | 21.0 |
| 5T | Tumor | UND | 28.18 | 25.9 | 26.8 | 27.2 |
| 1T-1 | Tumor | 31.6 | 23.56 | 17.4 | 26.3 | 19.6 |
| 1T-2 | Tumor | 29.8 | 23.63 | 18.7 | 26.1 | 20.0 |
| 6T | Tumor | 27.0 | 22.08 | 18.5 | UND | 20.0 |
| 7T | Tumor | 25.9 | 20.93 | 15.9 | 29.7 | 17.4 |
| 2T | Tumor | 28.3 | 21.37 | 19.0 | UND | 18.9 |
| 3T | Tumor | 27.9 | 21.73 | 16.9 | 32.6 | 18.8 |
| 8T | Tumor | UND | 24.06 | 20.3 | 29.9 | 20.9 |
| 9T | Tumor | 29.7 | 24.76 | 22.2 | 28.4 | 23.2 |
| 4T | Tumor | 28.9 | 20.59 | 19.6 | 31.6 | 20.1 |
| CTRL | Control Sample | 20.9 | 20.83 | 14.1 | 22.1 | 14.8 |

UND—Cq value undetermined; Control Sample—XpressRef Universal, Total RNA (QIAGEN) was used as a positive control for gene expression.

## 2.2. Conducting Microtomy and RNA Extraction

For the microtomy process, FFPE samples were cut using a rotary microtome (Capetown, South Africa) following the method of Sy [19]. Paraffin blocks were grouped into complete cases, excess wax was removed, and blocks were cooled on a cold plate for trimming. The knife blade was carefully inserted and positioned correctly, and the paraffin blocks were placed in the chuck holder. Trimming was performed at a thickness ranging from 10 to 30 μm, and the final tissue sections were cut at the designated thickness. The paraffin ribbon was carefully picked up and laid down onto a water bath, then smoothed with a paintbrush before being placed in nuclease-free tubes.

For RNA extraction, 400 μL of Deparaffinization Solution was added to samples, which were incubated at 55 °C for 1 min then vortexed briefly as per manufacturer's protocol. After removal of the solution, a mixture of DNase/RNase-Free Water, 2X Digestion Buffer, and Proteinase K was added. The samples were incubated at 55 °C for 1 h, then at

94 °C for 20 min. DNA/RNA Lysis Buffer was added, the tube was centrifuged, and the supernatant transferred to a Zymo-Spin$^{TM}$ IICR Column. Ethanol was added and the mixture transferred to a new column. After centrifugation, DNA/RNA Wash Buffer was added and the eluted RNA was stored at −80 °C.

### 2.3. Quantitative Reverse Transcription-Polymerase Chain Reaction (qRT-PCR)

This study aimed to analyze gene expression (of ABCB1 and CYP1B1) in RNA samples from FFPE tissues. Specific objectives included RNA quality control analysis, qPCR primer design, primer synthesis, cDNA synthesis, qPCR optimization and validation, and gene expression analysis. Primers were designed using Primer-Blast or sourced from the literature (Table 2), and qPCR efficiencies were assessed through standard curves. Assay specificity was confirmed using melt curves, and gene expression was analyzed using technical triplicates (Table 2). Reference gene stability was evaluated, with HSPCB and GAPDH selected for normalization. Preamplification uniformity and validation were conducted, and the experimental procedures covered primer design, preparation, cDNA synthesis, and preamplification. This study demonstrated successful qPCR assays and gene expression analysis in FFPE tissue-derived RNA samples.

### 2.4. Signaling Pathway Enrichment

To identify the signaling pathways and network associated with the expression of ABCB1 and CYP1B1, gene set enrichment analysis was conducted using the Enrichr online server (https://maayanlab.cloud/Enrichr/ accessed on 15 November 2023) [3,20]. The significance of these pathways was determined through *p*-value ranking, and the analysis included pathways sourced from the KEGG, WikiPathway, and PFOCR_Pathways databases.

### 2.5. Protein–Protein Interaction Network Analysis

To predict functional proteins that might contribute to the interactions and network of ABCB1 and CYP1B1, a protein–protein interaction network analysis was performed. This analysis utilized the STRING version 12 online server (https://string-db.org/ accessed on 15 November 2023) to unravel potential associations and functions within the identified proteins.

### 2.6. Molecular Docking Analysis

Following the administration of docetaxel in the patients' treatment regimen, we delved into an in silico exploration of the molecular interactions between the drug and the targeted genes, ABCB1 and CYP1B1, using molecular docking tools.

To conduct this in silico study, Molecular Operating Environment (MOE 2015.10) software was employed. Multiple 3D models for CYP1B1 (PDB ID 3PM0) were obtained from the Protein Data Bank website (PDB) (https://www.rcsb.org/ accessed on 19 November 2023) [21]. For ABCB1, amino acid sequences of the protein (P08183) were sourced from the Universal Protein Resource (UniProt) (https://www.uniprot.org/ accessed on 15 November 2023) and modeled using the Swiss Model online tools (https://swissmodel.expasy.org/ accessed on 15 November 2023).

Active ligands were generated employing 3D protonation, partial charge calculation, and energy reduction using Force Field MMFF94x. The proteins were created without adding repeat chains and water. MOE Quick Prep addressed structural defects, performed 3D protonation, and computed partial charge. MOE was then utilized to generate the optimal binding pocket under specific conditions, using a triangle matcher as the placement method and London dG as the major scoring function. An additional refinement step was executed using the rigid receptor technique with the GBVI/WSA dG score function to retain poses with the protein's most significant hydrophobic, ionic, and hydrogen-bond interactions. Subsequently, the compound–enzyme complex underwent visual analysis using BIOVIA Discovery Studio Visualizer.

### 2.7. Statistical Analysis

The RT-qPCR data underwent collection through a minimum of three independent experiments conducted in triplicate (n = 3). The qPCR analysis program (source, version) utilized was QuantStudio™ 12K Flex Software 1.5, subsequent analysis was carried out using Microsoft Excel SAS Version 9.2, and results were presented as means ± standard error of the mean (SEM). Statistical significance was determined using a one-sample *t*-test, with a *p*-value < 0.05 deemed statistically significant.

## 3. Results

### 3.1. Relative Gene Expression of ABCB1

The findings regarding ABCB1 expression levels in FFPE samples are illustrated in Figures 1 and 2. In Figure 1, ABCB1 expression is observed to be lower, but not statistically significantly, in prostate tumors compared to in normal prostate tissues. Figure 2 presents a comparative assessment of these samples, categorizing patients based on their responses to docetaxel treatment (good vs. poor responders). ABCB1, the gene responsible for encoding the MDR-1 ABC transporter, facilitates docetaxel transportation across the cell membrane. Upon contrasting good vs. poor responders in Figure 2, ABCB1 is notably and significantly expressed in poor responders.

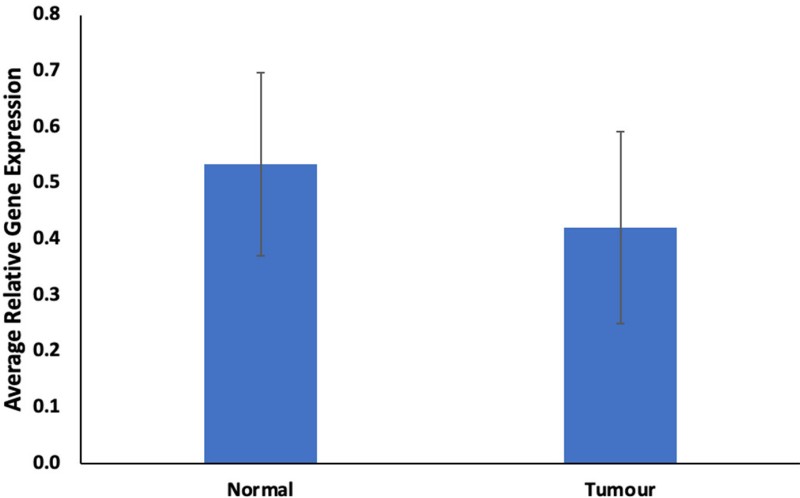

**Figure 1.** Expression of ABCB1 in prostate tumor. Values = mean ± SE; n = 3.

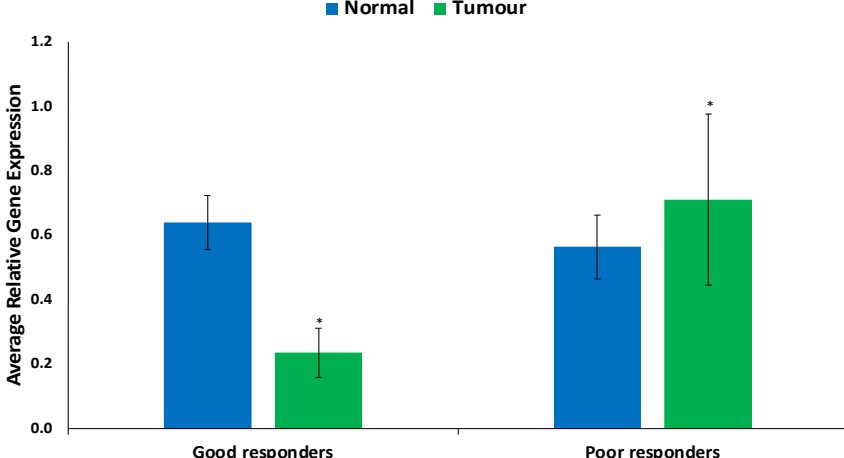

**Figure 2.** Expression of ABCB1 in prostate tumor of good and poor responders. Values = mean ± SE; n = 3. * Statistically significant (*p* < 0.05) in comparison to normal prostate tissues (normal).

### 3.2. Relative Gene Expression of CYP1B1

As shown in Figure 3, there was a significant ($p < 0.05$) upregulation of CYP1B1 in prostate tumors compared to normal prostate tissues.

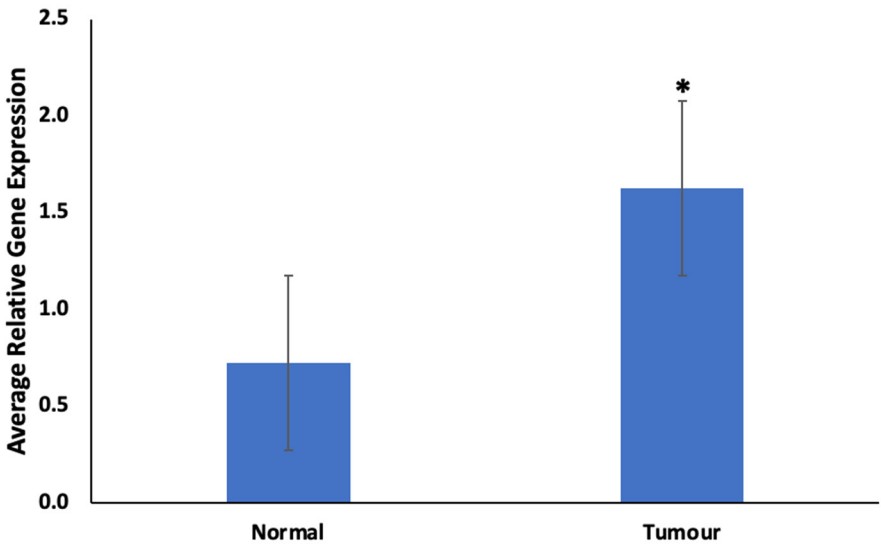

**Figure 3.** Expression of CYP1B1 in prostate tumors. Values are mean ± SE; n = 3. *—Statistically significant ($p < 0.05$) in comparison to normal prostate tissues (normal).

CYP1B1 was significantly ($p < 0.05$) overexpressed specifically in prostate tumors from poor responders versus good responders or compared to normal cells (Figure 4).

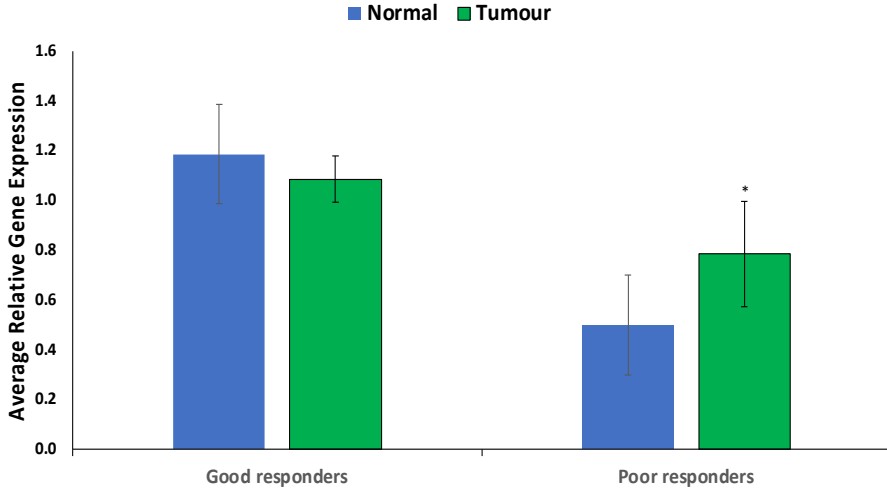

**Figure 4.** Expression of CYP1B1 in prostate tumors of good and poor responders. Values are mean ± SE; n = 3. *—Statistically significant ($p < 0.05$) in comparison to normal prostate tissues (normal).

### 3.3. Signaling Pathway Enrichment of CYP1B1 and ABCB1 Expression

Enrichment analysis of ABCB1 and CYP1B1 yielded a comprehensive set of 15 molecular functions and pathways, each associated with $p$-values < 0.05 as detailed in Table S3. As shown in Figure 5, the network map highlighted associations of ABCB1 with ABC transporters and LncRNA-mediated mechanisms of therapeutic resistance (WP3672). CYP1B1 was linked with ovarian steroidogenesis, tryptophan metabolism, steroid hormone biosynthesis, benzo(a)pyrene metabolism (WP696), the sulindac metabolic pathway (WP2542), and the estrogen receptor pathway (WP2881). Both ABCB1 and CYP1B1 exhibit associations with microRNAs in cancer and the Nuclear Receptors Meta-Pathway (WP2882).

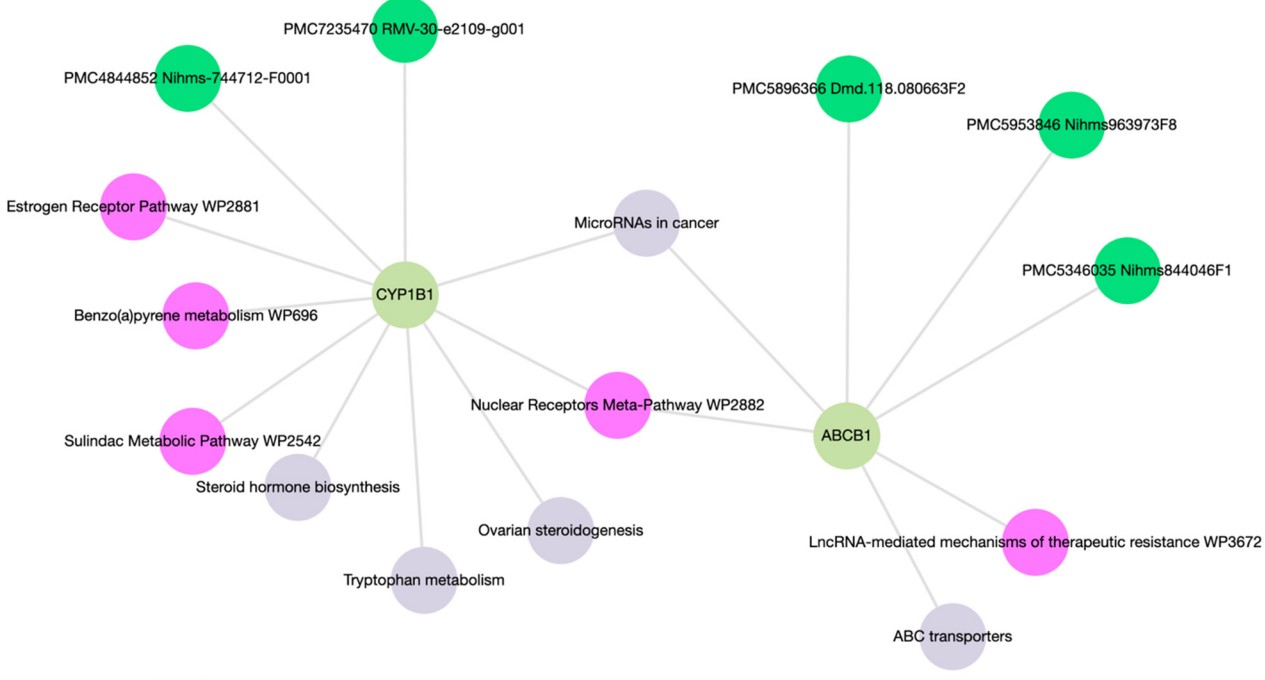

**Figure 5.** Network map of molecular functions and pathways following gene enrichment.

*3.4. Protein–Protein Interaction Network Analysis of CYP1B1 and ABCB1*

As shown in Figure 6, STRING analysis predicted protein–protein interactions of ABCB1 and CYP1B1 with Glutathione S-transferase Pi (GSTP1), Catechol O-methyltransferase (COMT), UDP-glucuronosyltransferase 1-6 (UGT1A6), Leucine-rich Transmembrane and O-methyltransferase (LRTOMT), and Epoxide hydrolase 1 (EPHX1), with 0.973, 0.971, 0.966, 0.966, and 0.966 scores, respectively (Table 3).

**Table 3.** Predicted functional proteins and scores.

| Protein | Score |
|---------|-------|
| GSTP1 | 0.973 |
| COMT | 0.971 |
| UGT1A6 | 0.966 |
| LRTOMT | 0.966 |
| EPHX1 | 0.966 |

GSTP1—Glutathione S-transferase Pi; COMT—Catechol O-methyltransferase; UGT1A6—UDP-glucuronosyltransferase 1-6; LRTOMT—Leucine-rich Transmembrane and O-methyltransferase; and EPHX1—Epoxide hydrolase 1.

*3.5. Molecular Docking Analysis of ABCB1 and CYP1B1*

Examining docetaxel through molecular docking analysis with ABCB1 and CYP1B1 unveiled robust molecular interactions of docetaxel with ABCB1 and CYP1B1 (Figures 7 and 8). Table 4 represents the free binding energies and types of hydrogen bonds between atoms of ligands and amino acids of the receptors. Docetaxel had the lowest binding energy with CYP1B1, with a binding energy of $-20.37$ Kcal/mol, and it had a binding energy of $-15.25$ Kcal/mol with ABCB1. Furthermore, docetaxel showed 3 H-donor and H-acceptor bonds linked to GLU874, MET875, GLN945, and TYR949 amino acids within the active site of ABCB1. In contrast, it displayed 2 H-acceptor and pi-H bonds connected to ILE399, PHE463, and SER331 amino acids in the pocket site of CY1B1 as shown in Figures 7 and 8.

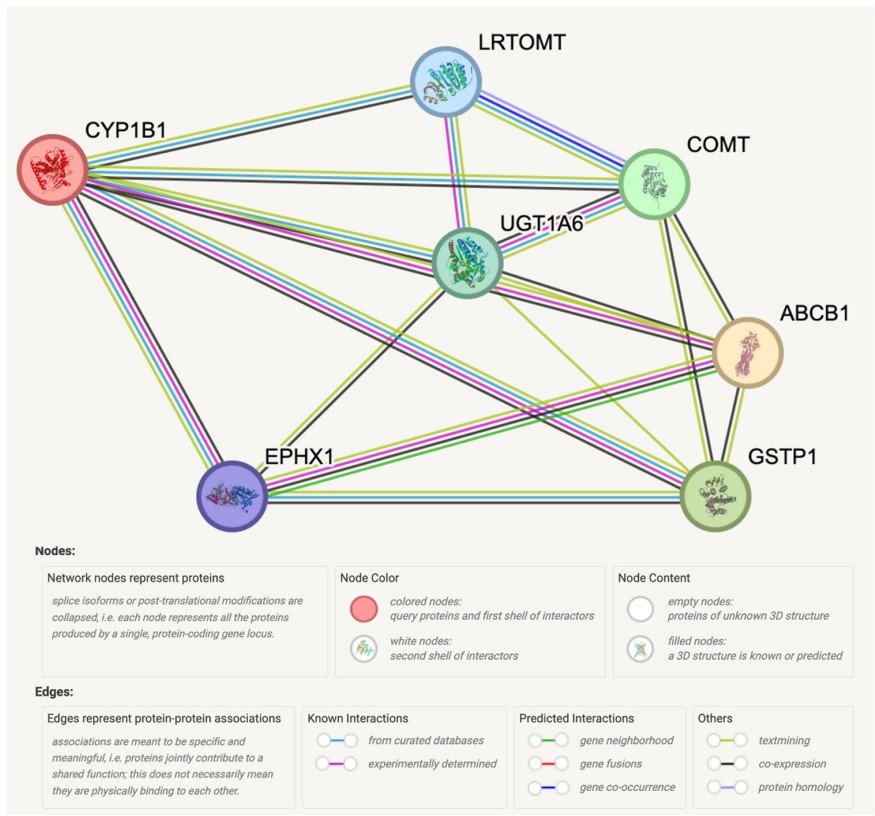

**Figure 6.** Protein–protein interaction network of CYP1B1 and ABCB1 genes with predicted functional proteins visualized by STRING.

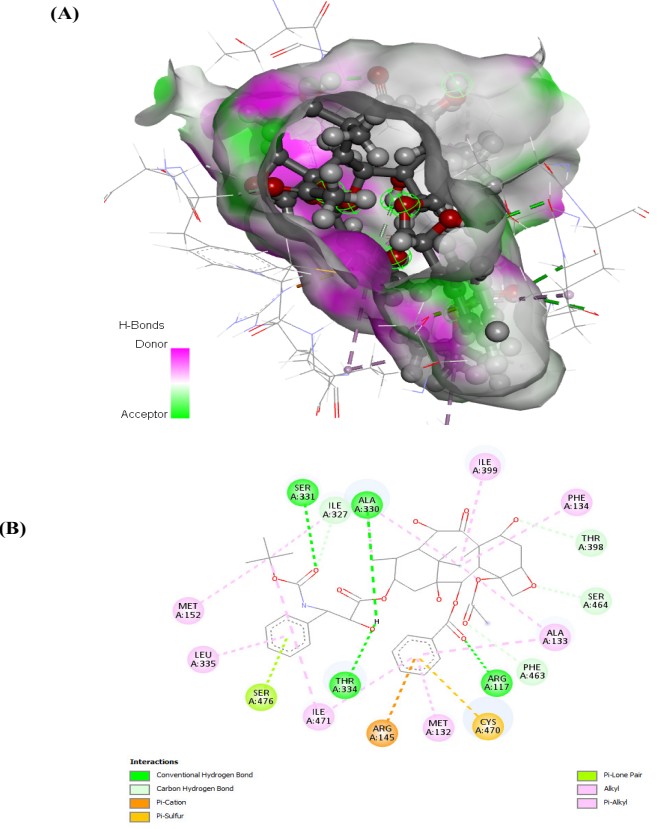

**Figure 7.** (**A**) 3D and (**B**) 2D representations of the molecular interaction of docetaxel with CYP1B1.

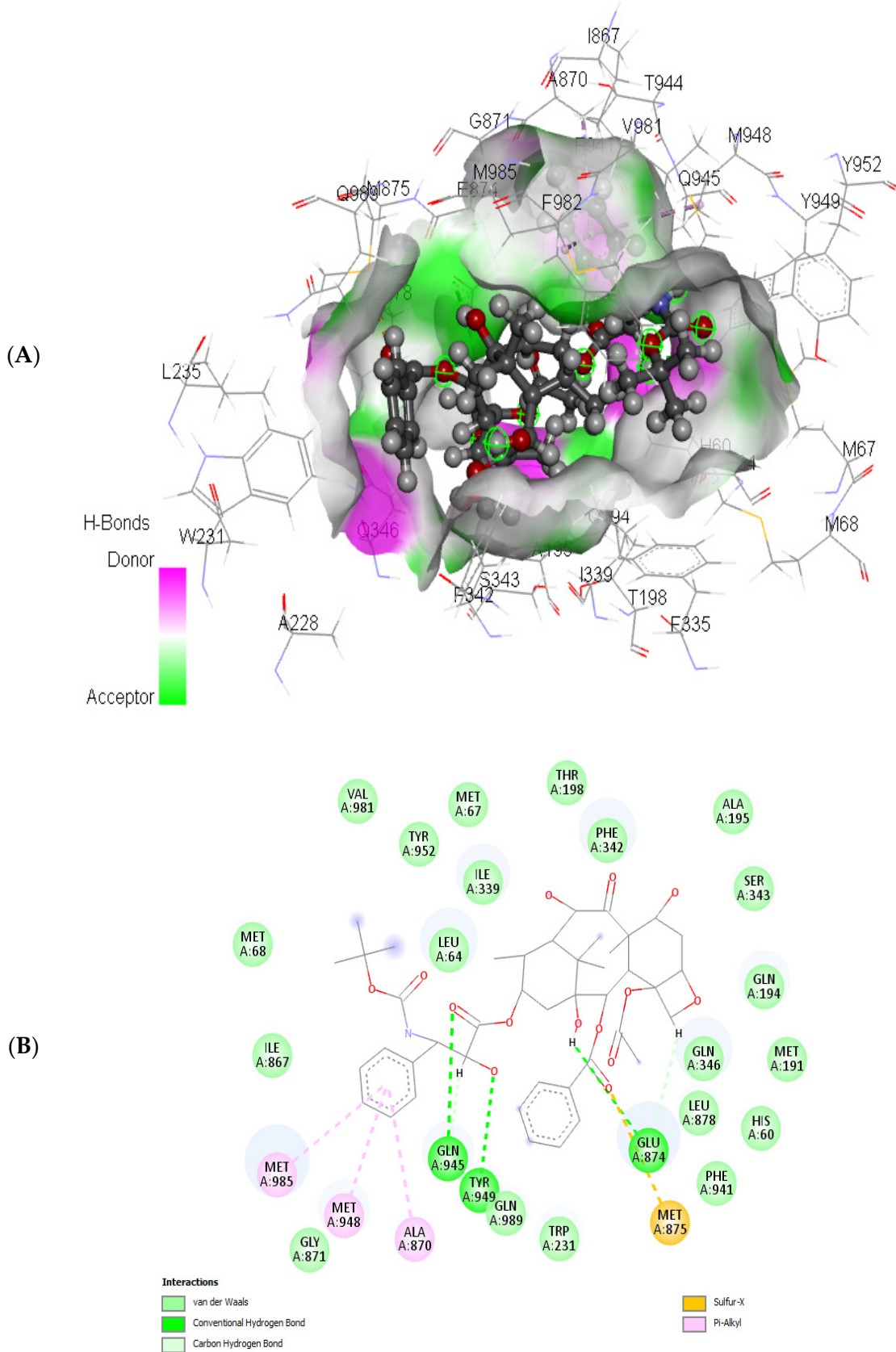

**Figure 8.** (**A**) 3D and (**B**) 2D representations of the molecular interaction of docetaxel with ABCB1.

**Table 4.** Free binding energy interaction of docetaxel with ABCB1 and CYP1B1.

| Ligand | Proteins | Hydrogen Bonds between Atoms of Ligands and Amino Acids of Receptor | | | | | | Score (Binding Energy) (Kcal/mol) |
|---|---|---|---|---|---|---|---|---|
| | | Ligand Atoms | Receptor | | Type | Distance (Å) | Energy (Kcal/mol) | |
| | | | Atoms | Residues | | | | |
| Docetaxel | ABCB1 | O4 | OE1 | GLU874 | H-donor | 2.92 | −1.2 | −15.25 |
| | | O10 | SD | MET875 | H-donor | 3.18 | −0.3 | |
| | | C41 | OE1 | GLN945 | H-donor | 3.37 | −0.9 | |
| | | O12 | OH | TYR949 | H-acceptor | 2.69 | −1.6 | |
| | CY1B1 | O5 | N | ILE399 | H-acceptor | 2.76 | −1.6 | −20.37 |
| | | O9 | CA | PHE463 | H-acceptor | 2.95 | −0.7 | |
| | | 6-ring | CA | SER331 | Pi-H | 3.92 | −0.8 | |

The interactive bonds were identified as hydrogen bonds (H-acceptor and pi-H) as shown in Table 4 and Figures 7 and 8.

## 4. Discussion

The escalating incidence of prostate cancer in sub-Saharan Africa, amid economic and healthcare challenges, is of grave concern. The conundrum deepens with rising drug resistance, intensifying the disease's impacts on morbidity and mortality. Despite years of extensive research and significant financial investment, there has not been a successful breakthrough in addressing prostate cancer. Each individual, and even each cancerous area within the same tumor, possesses a distinct transcriptome structure. These variances extend beyond mere gene expression levels to encompass the regulation and interaction of specific genes. The diverse and non-reproducible nature of transcriptomic organization among individuals renders the search for universal biomarkers and one-size-fits-all treatments impractical [22]. This study delves into the expression of drug resistance-associated proteins, namely ABCB1 and CYP1B1, in prostate cancer tumors and normal tissues. The categorization of patients based on their responses to docetaxel treatment—good responders (chemo-sensitive) and poor responders (chemo-resistant)—adds a nuanced perspective.

ABCB1, a member of the ABC transporters superfamily, plays a crucial role in drug transport and chemo-resistance, often exhibiting overexpression linked to multi-drug-resistant cancers [23]. In our study, low ABCB1 expression in prostate tumors hints at reduced chemotherapeutic resistance via the ABCB1 pathway (Figure 5 and Table S3). Gene enrichment of ABCB1 emphasizes this, highlighting suppressed drug resistance pathways (Figure 5 and Table S3). The intricate role of ABC transporters, crucial in drug transport, aligns with the observed low activation of ABC Transporter pathways in both normal and tumor tissues (Table 1). Strikingly, poor responders exhibit elevated ABCB1 expression (Figure 2), aligning with activated pathways identified in enrichment and network analyses (Figure 5 and Table S3).

CYP1B1, belonging to the cytochrome P450 enzyme family, holds significance in drug metabolism and is implicated in chemotherapeutic resistance across various malignancies. Elevated CYP1B1 expression in prostate tumors and particularly in poor responders (Figures 3 and 4) underscores a high potential for drug resistance. Pathways identified in enrichment and network analysis of CYP1B1 (Figure 5 and Table S3) align with its role in activating pathways associated with drug resistance in prostate cancer [24]. Unconventional steroid metabolism pathways driven by CYP1B1 are implicated in prostate cancer malignancy [25]. Aberrant tryptophan metabolism, linked to chemotherapy resistance via the aryl hydrocarbon receptor (AhR), further adds to the complexity [26,27].

Understanding resistance mechanisms holds the potential to facilitate the development of novel therapies for CRPC. The progress in advanced computational algorithms is particularly captivating as it offers opportunities to unravel the molecular intricacies underlying various cellular processes, such as protein interactions, molecular recognition,

mutation analysis, drug discovery, and bioengineering [28]. For instance, the predicted functional proteins from STRING analysis (Figure 6 and Table 3), including GSTP1, COMT, UGT1A6, LRTOMT, and EPHX1, are associated with drug resistance. GSTs, including GSTP1, contribute to chemotherapy resistance by detoxifying drugs [29]. UGT1A6's role in the metabolic inactivation of drug therapies is well-established [30]. EPHX1 overexpression is noted in castration-resistant prostate cancer [31].

The predicted free energies of −15.25 and −20.37 Kcal/mol (Table 4) suggests potent molecular interactions of the drug with ABCB1 and CYP1B1, respectively. This interaction can be attributed to the identified bonds, which bind the drug to the active site of the genes as shown in Figures 7 and 8. Previous studies have demonstrated increased survival rates in prostate cancer patients treated with docetaxel [32,33]. However, the therapeutic efficacy has been reported to be compromised in drug-resistant prostate cancer cells [34–36]. Furthermore, high expression of CYP1B1 has been implicated in the resistance of prostate cancer cells to docetaxel [25,37,38]. Multidrug resistance has also been reported in docetaxel therapy in prostate cancer cells with high expression of ABCB1 [39,40]. Thus, the potent molecular interaction of docetaxel with ABCB1 and CYP1B1 (Table 4 and Figures 7 and 8) may insinuate potential resistance of the cancer cells to docetaxel via increased expression of ABCB1 and CYP1B1. This correlates previous reports on increased expression of ABCB1 and CYP1B1 and cell proliferation following treatment with docetaxel [40,41].

## 5. Conclusions

In summary, our findings illuminate the vulnerability of cancer patients to drug resistance, which is particularly evident in the increased expression of ABCB1 and CYP1B1 in tumor samples from the poor-responders category, along with the associated molecular pathways. The strong molecular interaction observed between ABCB1, CYP1B1, and the chemotherapeutic drug docetaxel further underscores this susceptibility. While our initial study aimed to compare three patient categories—good responders, poor responders, and those with excessive toxicity—limitations in the available data restricted our analysis to only two groups (good and poor responders). The scarcity of patients exhibiting excessive toxicity in the South African government database posed a challenge. Additionally, our investigation focused solely on gene expression studies without delving into the identified pathways and predicted enzyme activities obtained from proteomics studies.

To deepen our understanding, we suggest conducting thorough research that examines these pathways and enzyme activities in both prostate cancer patients before and after treatment. This expanded scope will offer a clearer comprehension of how genetic factors influence drug transport and metabolism, contributing to the variability in individual responses to chemotherapy. Moreover, it will provide a more nuanced understanding of the potential molecular mechanisms underlying the drug's effects on patients. This pilot study serves as an initial exploration, establishing a proof of concept and methodological framework. However, it is crucial to emphasize the necessity of a more extensive investigation for comprehensive insights. Future studies should prioritize freshly obtained patient biopsies over archived specimens, ensuring an ample supply of RNA and proteins for precise expression level analyses. This approach will enhance the robustness and reliability of the findings, paving the way for more informed and targeted therapeutic strategies in the realm of prostate cancer treatment.

**Supplementary Materials:** The following supporting information can be downloaded at: https://www.mdpi.com/article/10.3390/cimb46030145/s1.

**Author Contributions:** Conceptualization, M.S. and L.G.; methodology, L.G., A.S. and O.L.E.; software, A.S. and O.L.E.; validation, L.G., M.S., B.A. and O.L.E.; formal analysis, L.G. and M.S.; investigation, L.G., B.A., A.S. and O.L.E.; resources, M.S., B.A. and S.M.; data curation, L.G., A.S. and O.L.E.; writing—original draft preparation, L.G.; writing—review and editing, M.S., B.A., A.S. and O.L.E.; visualization, M.S.; supervision, M.S., B.A. and S.M.; project administration, L.G., M.S. and

S.M.; funding acquisition, M.S. and S.M. All authors have read and agreed to the published version of the manuscript.

**Funding:** This research was funded by the National Research Foundation, grant number 129891 with Reference TTK200415513610.

**Institutional Review Board Statement:** The study was conducted in accordance with the Declaration of Helsinki, and approved by the Institutional Review Board (or Ethics Committee) of the Health Science Research Ethics Committee, Provincial Health Research Committee, and the National Health Laboratory Service, Bloemfontein, South Africa (Ethics Number: UFS-HSD2020/1278/2302-0004).

**Informed Consent Statement:** Patient consent was waived due to the anonymity of biological specimens, as stipulated by the National Ethics Guidelines: Department of Health Ethics in Health Research Guidelines, Processes, Structures, and Procedures (2015). The patients' specimens were handled in an anonymous manner, ensuring the absence of any of the patients' clinical information, and allowing for effective de-identification. The specimens were exclusively utilized for genetic and proteomic analyses, and the outcomes were associated with the de-identified specimens' treatment responses.

**Data Availability Statement:** All data are provided in the results and Supplementary of this article.

**Acknowledgments:** We extend our gratitude to A. Sherriff for accessing and identifying patients within good- and poor- responders' groups while maintaining anonymized information, and to L. Muller and his staff for their valuable assistance in the histopathological procedures.

**Conflicts of Interest:** The authors declare no conflicts of interest.

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
