# Peer review of "Genetic Signatures for Distinguishing Chemo-Sensitive from Chemo-Resistant Responders in Prostate Cancer Patients"

_cimb, doi:10.3390/cimb46030145_

Round 1

Reviewer 1 Report

Comments and Suggestions for Authors

Dear authors,

 Thanks for your contribution on this field. This is an interesting article aiming at characterizing drug resistance in prostate cancer.

Many points (see my comments in the pdf file) need to be solved to improve its potential before its acceptance for publication.

 All the best

Reviewer 2 Report

Comments and Suggestions for Authors
  • The paper may be vauable after incorporating some critical revisions. 
  •  
  • Minor problems: 
  • The study's conclusions might be strengthened by a larger, more diverse sample size.
  • While ABCB1 and CYP1B1 are important, the paper could explore a wider range of genes related to drug resistance.
  • The practical implications of these findings in clinical settings are not thoroughly discussed. Including potential strategies for integrating these genetic signatures into treatment planning could enhance the paper's relevance.
  •  

Major problems:

Table 2 and 4 should be moved to supplementary materials. 

In most figures caption "Values = mean SE; n = 3." seems very confusing. 

Figure 6 is very bad quality and needs to be prepared better. 

Table 5 is left with no explanation. 

Molecular docking results should be deeply compared with the regions of interactions with the other studies as docking is not the most precise option for such studies. Authors should discuss the limitations. 

Discussion is left with no deep-dive into the results nor with their comparison with other publications. 

Round 2

Reviewer 1 Report

Comments and Suggestions for Authors

Dear authors,

Thanks for considering my comments.

Now it is suitable for publicatoin

All the best

Reviewer 2 Report

Comments and Suggestions for Authors

The Authors have sufficiently revised the paper according to my comments.